# Computational Efficiency for the Surface Renewal Method

Jason Kelley and Chad Higgins

Dept. of Biological and Ecological Engineering, Oregon State University
116 Gilmore Hall, Corvallis, OR 97333 USA

*Correspondence to*: Jason Kelley (kelleyja@oregonstate.edu)

**Abstract.** Measuring surface fluxes using the surface renewal (SR) method requires programmatic algorithms for tabulation, algebraic calculation, and data quality control. A number of different methods have been published describing automated calibration of SR parameters. Because the SR method utilizes high frequency (10 Hz+) measurements, some steps in the flux calculation are computationally expensive, especially when automating SR to

perform many iterations of these calculations. Several new algorithms were written that perform the required calculations more efficiently and rapidly, and tested for sensitivity to length of flux averaging period, ability to measure over a large range of lag time scales, and for overall computational efficiency. These algorithms utilize signal processing techniques and algebraic simplifications that demonstrate simple modifications that dramatically improve computational efficiency. The results here complement efforts by other authors to standardize a robust and

accurate computational SR method. Increased speed of computation time grants flexibility to implementing the SR method, opening new avenues for SR to be used in research, for applied monitoring, and in novel field deployments.

## 1 Introduction

Originally described by Van Atta (1977), the SR model measures vertical flux that occurs during rapid events which manifest as coherent structures in a turbulent flow. The physical mechanisms are statistically distinct from those

described in the eddy covariance (EC) method, which has been established as a robust and accurate method to measure flux (Baldocchi, 2014). The surface renewal (SR) method offers several advantages and complements the use of EC to measure flux. While EC requires fast (10 Hz+) measurement of both the vertical wind speed and air temperature to measure the sensible heat flux, the SR method does not explicitly require vertical wind speed, allowing flux to be determined solely from rapid measurements of temperature or other scalar concentrations.

Because fewer, lower cost sensors are required, the SR method theoretically can be used for general applied monitoring (Paw U et al., 2005; Spano et al., 2000). Another advantage of SR is the ability to measure flux very near the surface or near the top of the plant canopy (Katul et al., 1996; Paw U et al., 1992). By taking measurements very close to the surface, the measurement fetch is reduced and the effective "flux footprint" is smaller (Castellví, 2012), yielding a more localized flux estimate.

The SR method estimates turbulent transport rates from fast response measurements of scalar properties such as temperature or trace gas concentration. In the SR conceptual model, rapid changes in scalar concentration are associated with episodic displacement of near-surface air parcels, and the surface condition is renewed from upper air. While in proximity to the surface, the air parcels are gradually enriched or depleted in temperature or scalar

concentration by diffusion (Castellví et al., 2002; Paw U et al., 1995). The majority of flux from the surface is attributed to these rapid ejections, which distinguish coherent structures in near surface atmospheric motions (Gao et al., 1989). The duration and amplitude of these rapid fluctuations (visible as ramps in the scalar trace) are used to determine the magnitude and direction of the flux density. Because of the short duration of these events, the SR method complements spectral methods to evaluate the flux contributions made over time scales shorter than the typical 15-30 minute averaging time used for EC (Katul et al., 2006; Shapland et al., 2012a, 2012b). Rapid flux measurement will facilitate new applications, such as spatial mapping of flux using vehicle mounted, near-surface sensors, and real-time monitoring systems. Mobile SR implementations and other novel field methods could provide new insights into the complexities of sub-basin scale hydrology, be used to validate downscaled models, and measure the heterogeneity of flux at sub-field scales.

The implementation of SR requires a prescribed averaging time period (on the order of minutes) and ramp time duration (on the order of seconds), for which a representative and statistically robust flux magnitude can be determined. To implement SR on a moving vehicle (for instance, to map spatially variable flux), finding a minimum averaging time is desirable to increase the spatial resolution of the resulting map. The averaging time and lag time used in the SR method relate the sensitivity of the scalar measurement to the time scales at which most significant flux occurs (Shapland et al., 2014). To find the minimum measurement period, field studies were conducted in 2014 and 2015 over various types of surface conditions. This required a rapid computational method that worked over a range of different time averaging periods, and which could implement the various calibration procedures used in the SR method. Initial attempts to calculate flux followed methods as described by Paw U et al. (2005a) and Snyder et al. (2008). However, implementing these methods as documented was hampered by slow computation time, which constrained the many iterations required to determine the minimum flux averaging period.

Open source software and online forums are abound with methods that utilize advances in computing power, memory availability, and the accessibility of multithreaded processing. These methods reduce computational overhead, and can augment the SR technique to allow implementation with low cost computers and data loggers, or where remote telemetry is required. Three example methods are shown here which streamline specific operational steps in the SR method. The first is a method adapted from signal processing to "despike" noisy data, a quality control technique commonly used in processing raw meteorological data. Second is a method to compute structure functions over multiple time lags rapidly using convolution in two dimensions. Third, an algebraic array calculation is used to find the cubic polynomials roots used to determine the SR ramp amplitude. By using more efficient algorithms, rapid iterative trials can be conducted to adjust calibration parameters, test hypotheses on the time averaging of flux calculations, and potentially measure SR flux in real-time.

Advantages such as low cost, relatively simple instrumentation, and easier field implementation are all cited as motivating factors to use the SR method (Paw U et al., 2005), yet work remains to standardize a robust method (French et al., 2012; Suvočarev et al., 2014). Because sensor cost is reduced, SR systems can be implemented to measure flux more extensively than EC, and in situations where EC is impractical. Extensive, site specific SR estimates can augment the utility of sparsely located, permanent weather stations in mapping the heterogeneity of surface flux. Examples of situations which could benefit from low cost flux measurements include direct crop ET

monitoring, experiments at remote field sites, and in developing regions. While SR may expand flux measurement applications, the method still requires standardized calibration and quality control measures to establish that SR is robust and accurate, and a critical step in developing the method is to reduce computation costs.

## 2 Methods

The example algorithms shown here improve or economize existing calculation methods, including despiking of time series data (Højstrup, 1993; Starkenburg et al., 2016), calculation of structure functions (Antonia and Van Atta, 1978), and Fourier analysis of signals i.e. spectral analysis (Press, 2007; Stull, 1988). In each case, dramatically faster execution times were accomplished using simple programming improvements. Most efficiency gains were a result of code vectorization, which is the conversion of iterative looping algorithms into array calculations. All

methods described here were implemented in the Matlab language (The Mathworks Inc., 2016), with the Statistics, Curve Fitting, and Signal Analysis Toolboxes. Matlab's Profiler (*profile.m*) was used to track the memory demand and time to implement calculations. Trials were conducted on multiple desktop systems; for uniformity, analysis shown here only used test runs that were conducted on a Windows 10 operating system running on an Intel Core (TM) i7-3720QM processor operating at 2.60GHz with 16GB RAM. Processor clock speed was verified using

Matlab's Profiler tool at run time, and processing times reported are described as run time (actual observed execution time) or as Total Run Time, which is the sum of CPU time for all calculation threads. Example methods are indicated by *function name in italics*. Abbreviated, commented scripts for the example functions are provided in the supplementary material. Example data was collected during various field experiments from 2014-2017, using an integrated sonic anemometer and infrared gas analyzer (IRGASON) and fine wire thermocouples (FWTs), and was

recorded at 10, 20, and 100 Hz using a CR1000 datalogger (Campbell Scientific). The data used in verifying the methods is provided through supplementary materials online (DOI: 10.7267/N9X34VDS).

Vector and array calculations are more efficiently executed than iterative methods; vectorization of Matlab code entails removing loops (which are not pre-compiled) and taking advantage of implicit parallel methods in Matlab's pre-compiled functions (Altman, 2015). Other significant improvements were enabled through the Fast Fourier

Transform by using convolution of number arrays, rather than iterative operations. In the case of determining ramp geometry in the SR method, Cardano's solution for depressed cubic polynomials (published in 1545) reduces a root finding algorithm from an iterative numerical approximation to an exact algebraic vector calculation. While some of the implementation of these methods are particular to the Matlab language, the general mathematical concepts are universal. Although these methods were prototyped in Matlab, the examples shown are generally useful as solutions

to challenges commonly encountered in micrometeorology.

### 2.1 Despiking of noisy data using convolution

Despiking is the removal of erroneous or extreme data points from a time series of sampled values. It is a common procedure when measuring environmental parameters, especially in challenging conditions or complex environments (Göckede et al., 2004; Starkenburg et al., 2016). The origin of spikes in a time series may be or

electronic or physical (sensor malfunction or actual physical non-errors). Regardless of the origin, spikes can be

recorded as abnormally large or small values, or may be marked by a firmware defined error flag. Spikes become problematic if they are not readily differentiated during automatic data imports (Rebmann et al., 2012). Spikes interfere with statistical calculations, and require some deliberate and objective method to identify, remove, and interpolate where they exist. For instance, a data logger program may record an error as "9999" or a character string, while Matlab denotes missing values in a numerical array as NaN ("not a number"). Because normally distributed data may contain noise in a wide range of values, robustness of the despiking algorithm is complicated by the requirement to differentiate between "hard spikes" characteristic of automatic flags (such as 9999) and "soft spikes" which are realistically valued but erroneous measurement. An objective limit for soft spikes is usually defined as appropriate for the signal to noise ratio of any particular data, usually in terms of variance during a defined windowing period. Clearly distinguishing errors can be achieved by a static objective criteria, by a dynamic statistic, or in a separate pre-processing operation. Previous authors have described a variety of methods including use of autocorrelation (Højstrup, 1993) and statistics within a moving window (Vickers and Mahrt, 1997). A comprehensive review of despiking methods is presented by Starkenburg et al., (2016), with emphasis on the accuracy and statistical robustness of different computation methods.

Despiking is a problem of conditional low pass filtering; consequently this procedure can be treated as an application of signal processing which can be performed efficiently using convolution. Mathematically, convolution can be understood as a multiplicative function that combines a data signal with a filter signal. For a discrete signal, the filter is a weight array which is multiplied (in the Fourier domain) with data inside a window. In the time domain, the window can be visualized as moving along the data array as it is multiplied. As examples, a filter with a weight of 2 at the center of the window, and zeros elsewhere, would amplify the data signal by a factor of two; a filter 10 samples wide, each weighted at 0.1, generates a running average of the data. The computational efficiency of convolution is a product of the Fast Fourier Transform (FFT), which allocates memory efficiently by a process known as bit switching. A thorough treatment of bit switching can be found in Chapters 12 and 13 of Press (2007). To demonstrate the increased efficiency of the FFT, two methods were used to despike 8.5 hours of 20 Hz sonic temperature data (609139 samples). One method utilized a *for* loop, following the objective criteria described by Vickers and Mahrt (1997). The second method (shown in *despike.m*) used convolution to determine a running mean and standard deviation used in the identification of spikes. After multiple runs with different input criteria, the first program average run time was 27 seconds. Using convolution, the second program average run time was 0.2 seconds, decreasing run time by approximately 99%. While this drastic improvement may potentially overemphasize slow compile times of *for* loops in Matlab (compared to other languages), it nonetheless demonstrates the value of the FFT in calculations with time domain signals. Faster processing time facilitates more comprehensive, calibrated, and accurate analysis, and can reduce data loss compared to coarser filtering techniques.

To test computation time uniformly, identical 10 Hz data was sub-sampled to record lengths of 0.25 to 48 hours, and multiple runs were despiked with each sample set. Raw data was checked for hard error flags which required text to number conversion, but was not otherwise manipulated prior to despiking. Matlab Profiler was used to track the run time for all threads, using the undocumented flag "built-in" to track pre-compiled Matlab functions as well as user

functions[1]. The total run time for all threads was tabulated and averaged across sets of each data length (Figure 1). By using convolution, despiking was two orders of magnitude faster for all lengths of data. To illustrate the effect of Matlab's built-in parallel processes, Figure 2 shows the ratio of actual run time to Total Run Time, indicating that the convolution method relies on computations conducted in parallel for processing increasingly longer data records. This benefit is direct accrued from the efficiency of the FFT.

With increased computation speed, automatic and accurate despiking can be accomplished, with reduced time cost to determine any necessary calibrate for the procedure. The various methods employed to despike data are variously limited by computational inefficiency (Starkenburg et al., 2016). "Phase space thresholding", originally described by Goring and Nikora (2002) is one such method that Starkenburg noted as being hampered by computational costs, and by a requirement for iterative applications to calibrate despiking parameters. By decreasing the execution time, a similar method was developed that allows rapid and accurate despiking of data, both for the detection of hard and soft spikes. A phase space method allows objective criteria to be calibrated for specific sensor data, and a visual diagnostic phase space diagram that allows for rapid calibration of the despiking criteria (Figure 3). Projecting the signal into a phase space diagram reveals modes related to sensor error, response time, and other factors leading to spikes. Using convolution to determine moving window statistics (such as a moving mean, standard deviation, etc.), objective identification of behaviors characteristic to a particular sensor response. In figure 3, infrared gas analyzer data (in this case, signal strength) collected at 20Hz for 17 days is projected with one minute moving window statistics. Based on this projection, a cut-off in phase space for spike identification can be assigned, and the subsequent percentage of removed data calculated. In this case, the sensor was repeatedly affected by dust from farm operations (Figure 4), yet only 1.5% of the data was required to be removed as spikes due to the precision of the despiking algorithm. This procedure took less than five seconds of computation.

**2.2 Structure function calculation**

Another computationally intensive process in SR is the determination of the 2nd, 3rd, and 5th order structure functions. Ramps are an identifiable feature in the measured temperature trace above any natural surface, yet determining the characteristic ramp geometry from high frequency data requires an efficient, robust, and preferably automated procedure. There are several methods to determine ramp geometry, including visual detection (Shaw and Gao, 1989), low pass filtering (Katul et al., 1996; Paw U et al., 1995), wavelet analysis (Gao and Li, 1993), and structure functions (Spano et al., 1997). Structure functions in particular provide both objective criteria to detect ramps and an efficient method to tabulate statistics of time series data, and use of structure functions has become the predominant method used for SR. The general form for a structure function is:

$$S^n(r) = \frac{1}{N-r} \sum_{1}^{i=N-r} \left[ T(i+r) - T(i) \right]^n \tag{1}$$

in which a vector of length **N-1** is composed of differences between sequential (Temperature) samples **T(i)**, separated by lag **r**. The structure function **S$^n$(r)** of order **n** for a given sample lag **r** is obtained by raising the

---

[1] http://undocumentedmatlab.com/blog/undocumented-profiler-options-part-4, Accessed January 2017

difference vector to the **n** power, summing the vector and normalizing by **N-1**. In a turbulent flow field, the sampled fluctuations of scalar time series **T(i)** are a combination of random fluctuations and coherent structures (Van Atta and Park, 1972). The random (incoherent) part of the signal is a product of isotropic turbulent processes, and over an adequately large sample this sample should have no particular directional sense or orientation (by the isotropic definition). On the other hand, coherent structures generate characteristic anisotropic signatures, with periods of gradually change punctuated by sharp transitions. These sharp transitions occur during "sweeps and ejections" of parcels enriched or depleted in scalar concentration (heat or water trace gas), evidence of transport from an Eulerian perspective. Structure functions can be used to decompose the time series fluctuations into isotropic and anisotropic components and identify the characteristic ramp amplitude and duration of coherent structures (Van Atta, 1977). Advances in sensor response time and processor speed have revealed an increasingly detailed picture of the coherent ramp structures. In deriving a method to find ramp geometry, Van Atta (1977) calculated structure functions for eight different lags. Two decades later, increased processor power and memory size allowed Snyder et al., (1996) to calculate structure functions on 8 Hz data for lags from 0.25-1.0 seconds, but they were unable to resolve fluxes accurately at some measurement heights and surface roughness conditions. Later it was realized that determining the contributions from "imperfect ramp geometry" would require more thorough examination of ramp durations (Chen et al., 1997a; Paw U et al., 2005).

For this analysis, data was used from several field experiments. The data records used ranged in length from 8 hours to over two months, with sampling frequencies of 10Hz, 20Hz, and 100Hz (fastest frequency for short duration trials only). Initially, computation of structure functions with the first method (series of nested *for* loops) for 3 minute periods with lags up to 10 seconds required an average 39 second computation time. In contrast, using the convolution method, this same calculation was accomplished in 7.6 seconds, an ~80% reduction in execution time. The function *strfnc.m* (provided in *Supplementary Materials, S1*) also simultaneously time stamps the averaging period, finds the sign of $S^3(r)$ (used to find flux direction), and indexes the maximized value of $S^3(r)/r$, preparing the data for subsequent steps in determining flux. Using 100 Hz FWT data increased processing time using the convolution method to 38.4 seconds. The loop method would be unable to process 100 Hz data in real time applications, and would require long calculation time when using large continuous data records.

For a total of N sample lags, two dimensional convolution is performed using a filter matrix which is composed of N column vectors of length N+1: [1 -1 0 0...0], [1 0 -1 0 ...0],...[1 0 0 0.... -1]. Each column represents a sample lag increasing distance. When the filter matrix is convolved with time series data, the column vectors of the resulting matrix are vectors of the element-wise differences (T(i+r) - T(i)) as in Eq. 1; these vectors correspond to each sample lag in the filter. Trials of 10Hz data using Matlab's Profiler showed that calculation efficiency is not accrued directly from convolution, but by changing the order of implementation. In the looping method, exponentiation (n = 2,3,5) is conducted on the difference vectors for each lag separately. The accelerated exponentiation in *strfnc.m* is possible by using matrix multiplication on the convolved matrix, and is faster due to compact memory allocation of the FFT. The resulting efficiency (calculation time for a given data size) doesn't depend on total data size, but is strongly dependent on the length of the averaging period used to partition the data (Figure 5). In other words, the choice of averaging period length is the most significant factor in computation time of the maximized structure

functions used to determine ramp geometry. Computation time increases rapidly for periods shorter than 5 minutes. The length of averaging time is a critical consideration in developing a rapid SR measurement method.

In most SR studies to determine flux, a lag time is assigned to the structure function calculation, with only a few authors allowing for a procedure to maximize the ratio $S^3(r)/r$ (Shapland et al., 2014). Yet lag time has been
identified as a critical parameter in the linear calibration of ramp geometry to calculate flux (French et al., 2012). Because the ideal SR calculation identifies the lag which maximizes the ratio $S^3(r)/r$, the *strfnc.m* procedure calculates structure functions for a continuous range of lags up to an assigned maximum lag. Based on repeated trials over a broad range of stability conditions, a short maximum lag (3-5 seconds) is usually adequate under unstable conditions. Following the model of parcel residence time, this is likely a result of buoyancy and higher flux
magnitude leading to shorter ramp duration. Under stable conditions, though, longer lags are required to detect the true maximum of the ratio $S^3(r)/r$, indicating that the time scale contributing to flux increases. To evaluate sensitivity to the maximum calculated lag, the structure functions were calculated iteratively, varying the averaging period and maximum tested lag time (Figure 6). Regardless of the length of the assigned range of lags, the convolution method was between 4 and 14 times faster than the loop method, with short averaging periods again the
largest factor in the difference between the two methods. Using the 2-d convolution, automated selection of a lag in a continuous time series is feasible.

### 2.3 Cardano's method for depressed cubic polynomials

For the idealized SR method, the structure functions are retained (for each averaging period) for the lag which maximizes the ratio $S^3(r)/r$ - this lag is associated with the theoretical maximum contributing scale of flux. The
resulting values are used as coefficients in a cubic polynomial, the root of which is the ramp amplitude (**A**) used to calculate flux:

$$A^3 + \left(10S^2(r) - \frac{S^5(r)}{S^3(r)}\right)A + 10S^3(r) = 0 \qquad (2)$$

The magnitude of the real root (of 3 possible roots) is the characteristic ramp amplitude of the scalar trace (Spano et al., 1997). The Matlab root finding algorithm computes eigenvalues of a companion matrix to approximate the solution to a $n^{th}$ order polynomial, regarding the input function as a vector with n+1 elements (*roots.m*
documentation[2]). Consequently, this function cannot be executed directly on an array. On the other hand, an algebraic solution method can be applied to vectors. An appropriate method for this type of cubic polynomial was found by Gerolamo Cardano in the 1545 *Ars Magna*. Cardano's solution for "depressed" cubics (with no squared term) is found by substituting A with $(m^{1/3} + n^{1/3})$ into the abbreviated equation $A^3+pA+q=0$. Expanding terms and using the quadratic equation yields an exact solution:

---

[2] http://www.mathworks.com/help/matlab/ref/roots.html, Accessed 9AUG2016

$$A = \left( -\frac{q}{2} + \sqrt{\left(\frac{q}{2}\right)^2 + \left(\frac{p}{3}\right)^3} \right)^{\frac{1}{3}} + \left( -\frac{q}{2} - \sqrt{\left(\frac{q}{2}\right)^2 + \left(\frac{p}{3}\right)^3} \right)^{\frac{1}{3}} \tag{3}$$

where **p** and **q** are coefficients in the depressed cubic and derived from the structure functions (Edwards and Beaver, 2015). The function *cardanos.m* was adapted from a function by Bruno Luong[3], in a reduced form for the real valued cases used to implement the SR method. The function output was verified against the Matlab function *roots.m* for polynomials with both positive and negative real valued inputs (imaginary inputs are applicable to ramp parameters). Solution for the real roots in this manner expedites determination of flux magnitude and direction. The

algebraic root finding method simplifies and speeds iterative application of the SR method by operating directly on arrays.

Solving for the roots of this function yields a single, predominant ramp amplitude from a given temperature trace (with units of °C or K). In addition to ramp amplitude, the time scale or ramp duration must also be determined. (Van Atta, 1977) suggested that ramp time $\tau$ should be related linearly to amplitude **A**, and proposed:

$$\tau = \frac{-A^3 r}{S^3(r)} \tag{4}$$

In practice, determination of ramp time $\tau$ from **A** using this equation requires an empirical calibration; this calibration has been shown to be related to surface conditions and instrumentation (Chen et al., 1997b; Shapland et al., 2014). Ongoing work using replicate measurements at multiple heights (Castellvi, 2004) and frequency response calibration (Shapland et al., 2014; Suvočarev et al., 2014) have begun to resolve the causes of variability in this parameter. In this study, it was found that the ratio in equation 4 remains essentially constant for a given surface

roughness condition, allowing determination of $\tau$ algebraically. Automated computation of equation 2 using the exact solution facilitates rapid evaluation of the ramp geometry, and determining flux magnitude from ramp geometry is a relatively simple matter of linear scaling when calibrating to a control measure such as eddy covariance.

### 3 Conclusions

As with other methods for measuring flux from the surface, analytic solutions do not always translate easily into straightforward numerical computation, especially when working with large data records or when calculating in real-time. In applied research, custom algorithms are often developed by individual researchers, requiring special training in programming, significant time investment, and the motivation to use sophisticated techniques that fully utilize available memory and processing power. Efforts to standardize the eddy covariance method (Aubinet et al.,

2012; Baldocchi, 2014) and data quality control (Allen et al., 2011; Foken et al., 2012) have not yet been similarly applied to the SR method, although substantial work has been made to validate and unify SR methods (Castellví, 2012; Chen et al., 1997b; French et al., 2012; Suvočarev et al., 2014). By appropriating methods common in signal

---

[3] https://www.mathworks.com/matlabcentral/newsreader/view_thread/165013, Accessed 10MAY2016

processing, and by sharing open source tools using online forums such as stackexchange.com, more sophisticated approaches can be implemented. In particular, reducing the computational overhead of calculating flux enables

broad implementation and robust verification of the SR method. Rapid algorithms allow for automated assignment of lag time, rather than fixed assignment, and allow flux determinations while varying the length of flux averaging periods. These procedures allow for comprehensive analysis of both the physical time scales of surfaces flux, and the sensor response and uncertainty associated with the SR derived flux. Calibration of despiking criteria can be implemented quickly at low computational cost. In summary, efficient methods for computing SR flux allow

implementation in novel deployments such as low cost, continuous monitoring, and on moving platforms. Future work remains to transfer efficient methods from the Matlab development platform to open source implementations, and enable hardware to perform these techniques directly for real-time applications. Reducing the cost and power requirement of the required data loggers, computers, and telemetry will facilitate the extensive deployment of SR sensors to aid in describing the heterogeneity of flux across the landscape.

**4 Data Availability**

All data used in this analysis and scripts implementing the algorithms described above are available online at http://hdl.handle.net/1957/60599. This supplementary materials is also listed with DOI: 10.7267/N9X34VDS Abbreviated scripts for the three example methods may be found in the supplemental materials. Requests for phase-space despiking methods can be directed to the corresponding author.

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

**Figures**

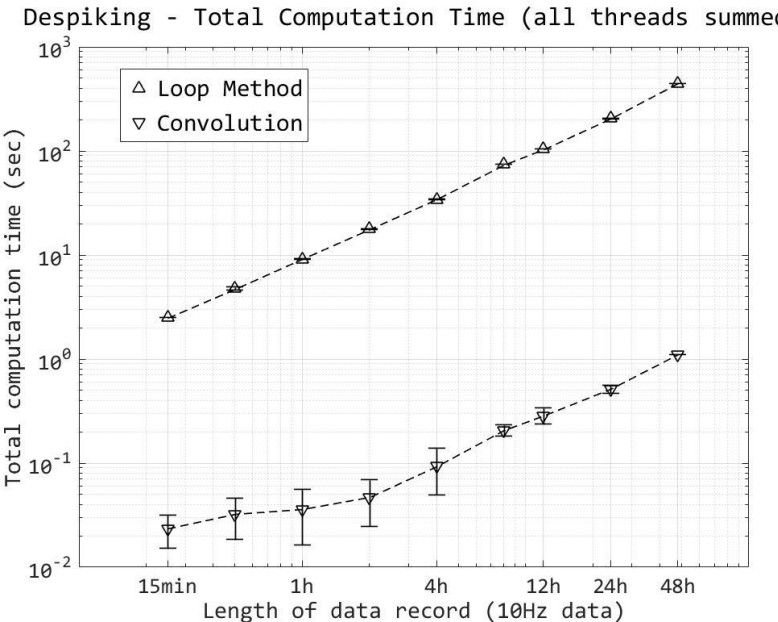

**Figure 1: The total computation time is the sum of CPU time spent on all calculation threads. Marker is the mean run time for multiple runs, which varied from 30 runs (15min - 4h data) to 10 runs (8, 12, 24 h). 48h calculation is represented by one run only. Error bars represent one standard deviation of all runs.**

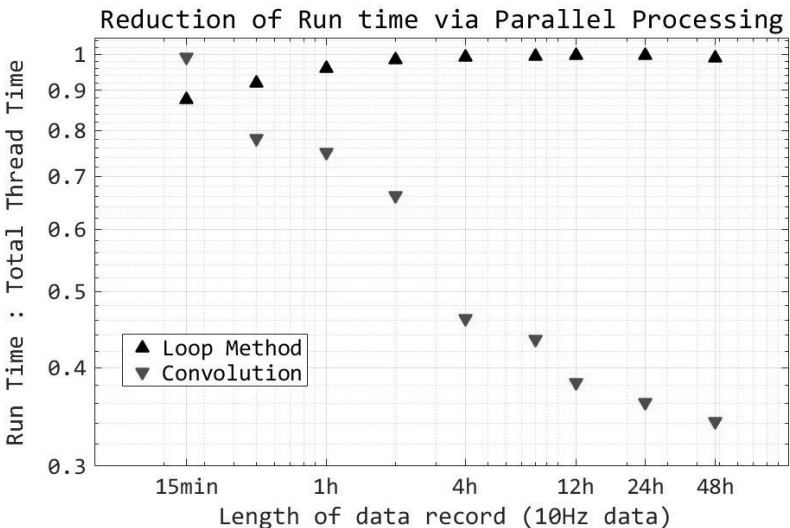

**Figure 2: Fast calculation of larger data sets is due to implicit parallel processing via the FFT, which is readily performed by multiple simultaneous threads. The efficiency of parallel processing is shown by a lower ratio of Run Time to Total Thread Time.**

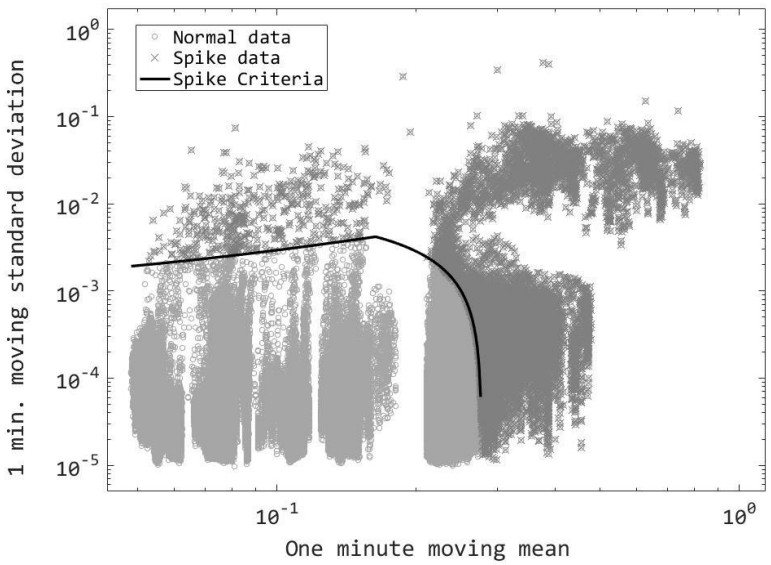

**Figure 3: Phase space diagram showing moving window statistics of IRGA signal (17 days of 20 Hz data)**

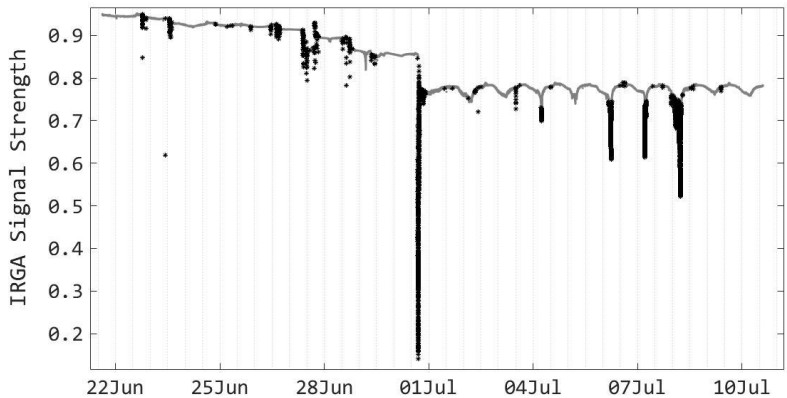

**Figure 4: Time series showing data removed as spikes (bolded) from phase space criteria in Figure 3.**

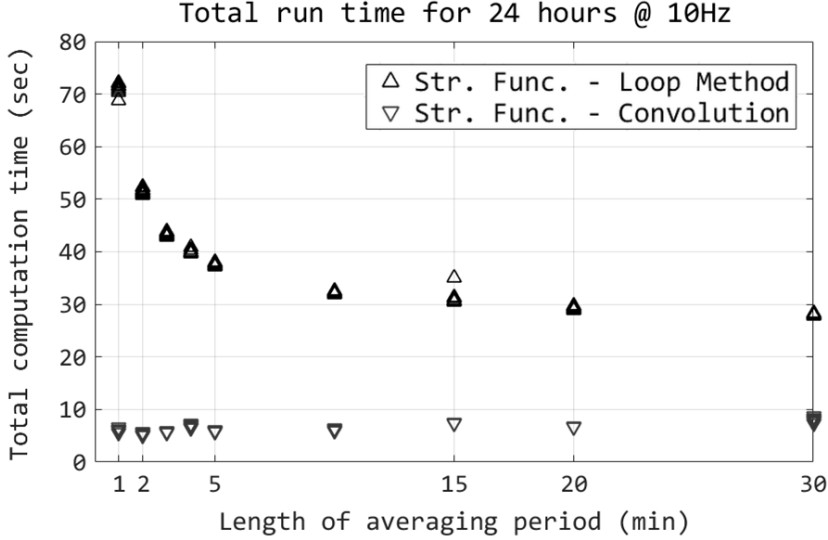

**Figure 5: Ten iterations of the structure function calculations using a range of averaging periods.**

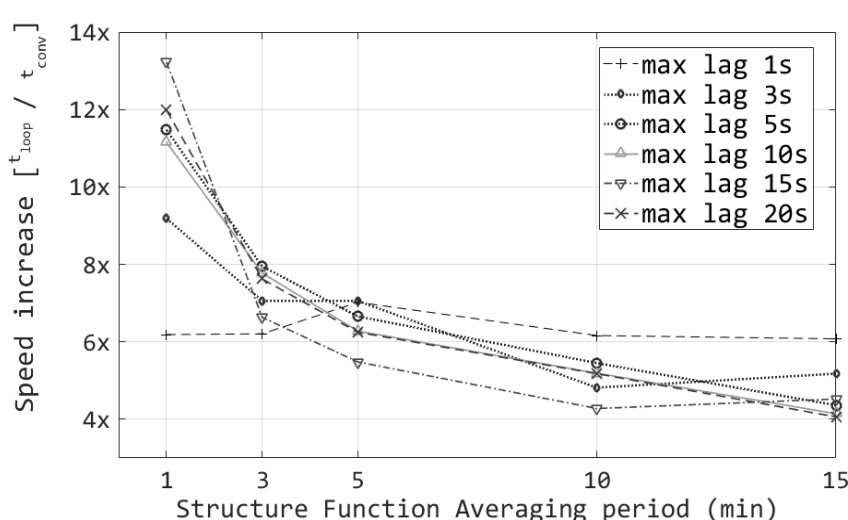

**Figure 6: The performance gains using convolution are more significant for short averaging periods, regardless of maximum lag used in calculating structure functions.**