# Peer review of "Computational Efficiency for the Surface Renewal Method"

_Atmospheric Measurement Techniques, 2017_

## Referee Comment (RC1) · Anonymous Referee #2 · 28 Jun 2017

Review of the article:

**Computational Efficiency for the Surface Renewal Method**

by Jason Kelley and Chad Higgins

GENERAL COMMENTS:

The article is well into scope of the AMT journal and I support its publication after the corrections suggested below. It describes useful toolbox as a new idea for the surface renewal method and the effort to make it more computationally efficient is important for micrometeorological community. The writing should be improved for the sake of clarity. The title should be changed to reflect the manuscripts idea of improving the initial part of the surface renewal method – the ramp detection method. It should be clarified further in the manuscript that the fluxes will be final once the Two-Scale Scalar Ramps method (Shapland et al., 2012a; 2012b) is used or appropriate calibration approach is implemented (i.e. different works by Snyder and/or Castellví). These articles should be referenced correspondingly. The Conclusions section should be rewritten completely. The content is more appropriate for the introduction section and lacks to show direct conclusions of the manuscript's purpose.

SPECIFIC COMMENTS:

P1L9: "20 Hz+" I suggest changing to "10+ Hz" since there are articles published on successful SR application over data collected at 10 Hz frequency. Similarly, it is mentioned 10+ Hz in the introduction section.

P1L13: Please add, "computational" before the word "efficiency"

P1L14: Please, be more specific, i.e. avoid saying "Programming techniques such as these", it is still not clear in the abstract what you are referring to. Also, "can grant" is not suitable to the rest of the sentence, since what you are stating is not the subject of your research presented in this manuscript, but just a suggestion. Try using "may be useful" or similar instead.

P1L19: Please add word "possible" between "determinations" and "using".

P2L50: Please clarify. It is said in this line that the convolution is used for structure functions computation, while later is said the convolution is used to despike the data. It seems that the convolution is used for both and should be explained.

P2L51: Replace "simplifying" with "to simplify"

P2L53: Please, start the Methods differently, i.e. avoid saying "These algorithms" since the algorithms are not very well defined so far in the manuscript and it is always better for the reader to be more specific.

P2L59: Add, "the method" before "implementation"

P4L105: Please change the word "approach" to "program" or "program run" for clarity.

P4L111-113: Please rewrite for the clarity.

P4L114: Please replace "identical" by "from the same 20 Hz dataset".

P4L131 and L133: Should "N-1" be actually "N-r"?

P4L133: Should "T(t)" be "T(i)" in accordance to Eq.1. and for clarity?

P5L137: "sweeps and ejections" of what? It should be clarified adding more description of the surface renewal method background.

P5L138: Please add "ramps in the temperature signal caused by" between "geometry of" and "coherent structures"

P5L140 and L141: Please change to be clearer that the detection of the structure functions in the scalar signals improved.

P5L144: Explain under which conditions.

P5L146: Please replace "to" by "for" if I understand well. In addition, explain why are the periods of 8 hours collected, and not continuously for two months. What challenges did you find? Was it only data collection for the unstable conditions?

P5L147-L150: Please explain how this analysis is useful. Are fluxes better to be calculated for 3 minute periods?

P5L163: What "total data size" means? It works with the same efficiency over 20Hz and 100Hz dataset? Please clarify.

P5L169-170: Is this the result of the authors' own research? If not, please give a reference.

P6L174: "again the largest factor in the difference between two methods" is making this phrase unclear. Please change to clarify.

TECHNICAL CORRECTIONS:

P1L10: One extra space seems to be typed in between the words "demonstrate" and "that"

P1L26: I think "manifests" is necessary instead of "manifest"

P2L60: "on" instead of "one"

P3L95: "an" instead of "a" before "application"

P4L111: Please use different word instead of "conditioning" if possible (i.e. "despiking")

P4L135: Please replace "product of" instead "product by"?

P4L135: Please replace "sense" by "importance in the flux measurement" or similar. Also, replace "generate" by "are"

P5L158: Please replace "difference vectors" by "the vectors of the differences". In addition, change "for each sample lag" by "for the corresponding sample lag"

P5L165 and 166: are empty.

Avoid "@" in Figure titles.

---

## Referee Comment (RC2) · Anonymous Referee #3 · 7 Jan 2018

The authors present three methods to improve computational speed of sensible heat fluxes from surface renewal measurements. The methods aim to improve practicality of the SR approach. The manuscript is original research, technically correct, and appropriate for AMT. However, construction of the manuscript has problems. Lack of innovation is one: none of the proposed methods (de-spiking, FFT solution of lag, and root solving) are, though you could say that your implementation is novel. The abstract does not adequately present the problem being solved and their proposed solutions. The presentation here and in Conclusions mention different points but there needs to be a unified message. There is a lack of quantification supporting assertions in the Introduction and details in their tests in Methods. Current relevant references are not included.

[Figure]

Details: Abstract: Logical presentation in the abstract needs improvement. Sentence one is a factual and not very informative statement (don't all flux techniques require programmatic algorithms?). Sentence two addresses a problem never posed, but should be: existing knowledge on required averaging periods is incomplete or poorly understood; or maybe the problem is more specific: SR is computationally expensive and reducing the averaging period would greatly improve feasibility of SR. But then in sentences 4,5, and 6 it appears that discovering the minimum averaging period isn't really what your study is about, instead it is to develop computationally faster processing algorithms. Or maybe the intent is to facilitate mobile flux platforms as mentioned in the Introduction. Looking at your conclusions, a completely different idea emerges: the need to standardize SR methods. Why is this mentioned as an after-thought? What is your thesis?

Introduction: Not mentioned, but should be, are the SR limitations relative to EC: it returns H and not LE, the latter having to be computed by residual (and thus ET estimates contaminated with both SR-derived H and non-SR (G,Rn) errors. What instrument do you propose to deploy for SR? Thermocouples are inexpensive but fragile, 2D sonics not fragile but not inexpensive. So: would help readers to quantify the benefits: how much less does SR cost? What fetch benefit does one gain with SR? I think the jury is still out on SR fetch length, but estimates would still be important to support your claim. L43: a major problem here: you have yet to publish your manuscript assessing efficacy of different averaging periods, but as of now we don't know the result and so do not know how important your algorithm refinement is. L 51-56: here your actual manuscript content is presented: you implement 3 algorithms to speed SR processing: de-spiking, structure function computations, and a root solver. Regarding de-spiking: authors should read and consider findings in Starkenburg et al. 2016, with attention to their Table 1. Discussion of convolution should mention that the time-series operation includes multiplication of the time-reversed kernel.Regarding structure function lag estimation, consider findings reported in French et al. discussing lag vs. SR accuracy. Root solving is not innovative although I will grant innovation in the diagnostic findings

of pathological cubic equations by Edwards and Beaver. It would be useful to compare the speed improvement using Cardano's method vs. numerical Newton-Raphson. If your aim is open-source you should provide readers with evidence that m script successfully executes on a non-proprietary software platform. Line 166: here is a critical finding about computation time vs. averaging period; you should find a way to highlight this and not

---

## Author Response (AR1)

From reviewer #1, "Because the paper focuses on determination of ramp dimensions using VA method, this Ms must.. say something about the ramp period. Otherwise, a reader has the feeling that the Ms is incomplete.... the present Ms must to add a sentence about the ramp period. In fact, I do not understand why it was not included because after determination of the ramp amplitude and the time lag that peaks S3r/r the calculation of the ramp period is straightforward using VA. Thus, no extra figures or calculations must be implemented." I have added a more complete description of the SR calculation of flux, including the determination of the ramp period (tau) as it relates to computation efficiency. I have also reordered some of the method description to be more pertinent to the surface renewal method in particular.

Also from Reviewer #1, "Because the authors mentioned that there is a companion paper where they compared (I guess) SR and EC fluxes, one may hesitate the latter issue was addressed in the forthcoming Ms. I have clarified that the experiment alluded to in the manuscript does calibrate surface renewal flux on the basis of fluxes calculated via eddy covariance. From Reviewer #2," *I am eager to try the supplemental code over my own data.*" I have updated the manuscript with a declaration of data availability (with a DOI), and listed the URL from which data and complete code can be downloaded. *"Some clarification will be necessary in parts of the manuscript."* I have made some attempt to identify some ambiguous references and clarify these, expecting that the discussion forum will raise additional needed clarifications. *"Also, very important comment on possibility to use this computationally efficient approach in dataloggers programming for online flux computation is needed in more detailed discussion."*. While this is a research interest that will be pursued, no attempt has been made to date to implement such computational approaches in a data logger programming language. Nonetheless, it should be straightforward to program the most relevant aspects of these techniques in languages that are pre-compiled with methods such as the FFT (for example, Campbell Scientific loggers using CRBasic). The authors will attempt to address this in the discussion forum, as it is not a task that has actually been completed to date.

**GENERAL COMMENTS:**

The authors gratefully acknowledge the reviewer's comments, which have led to substantial changes in the manuscript, which was initially revised in May 2017, and again recently in response to additional review. Overall, I have attempted to improve clarity of writing; specifically, the hypotheses and goals of the manuscript have been clarified, and the methods are more clearly delineated with appropriate and up to date citations. The brief conclusion has been changed to be more appropriate, in particular to describe additional results which have been included in response to additional reviewer requests. The most significant of these was expanding on actual implementation of despiking via convolution, using a phase space criteria for quantitative spike identification. I have also attempted to clarify those parts of the analysis which are novel in implementation, within the context of previous work in programmatic algorithms.

Specific comments are addressed in the supplementary notes, attached.

**SPECIFIC COMMENTS:**

P1L9: "20 Hz+" I suggest changing to "10+ Hz" since there are articles published on successful SR application over data collected at 10 Hz frequency. Similarly, it is mentioned 10+ Hz in the introduction section.

Changed as requested. Further clarifications later in the text as to different sampling frequencies and the corresponding cost in data storage and computational efficiency.

*P1L13: Please add, "computational" before the word "efficiency"* Added as requested.

P1L14: Please, be more specific, i.e. avoid saying "Programming techniques such as these", it is still not clear in the abstract what you are referring to...

This section has been re-written to be more concrete and an appropriate summary.

*P1L19: Please add word "possible" between "determinations" and "using".* Added as requested.

P2L50: Please clarify. It is said in this line that the convolution is used for structure functions computation, while later is said the convolution is used to despike the data. It seems that the convolution is used for both and should be explained. Added clarification here as requested.

*P2L51: Replace "simplifying" with "to simplify"* Replaced as requested.

P2L53: Please, start the Methods differently... it is always better for the reader to be more specific.

Changed with the intent to be more specific for the reader.

*P2L59: Add, "the method" before "implementation"* Added as requested.

Atmos. Meas. Tech. Discussion doi:10.5194/amt-2017-123

*P4L105: Please change the word "approach" to "program" or "program run" for clarity.* Changed as requested.

*P4L111-113: Please rewrite for the clarity.* Changed as requested.

*P4L114: Please replace "identical" by "from the same 20 Hz dataset".* Replaced as requested.

*P4L131 and L133: Should "N-1" be actually "N-r"?* Yes, thank you for catching this error in the text.

*P4L133: Should "T(t)" be "T(i)" in accordance to Eq.1. and for clarity?* No, this sentence refers to the time series rather than the discrete samples, but I agree it was unclear due to proximity of the terms. It has been changed for clarity.

*P5L137: "sweeps and ejections" of what? It should be clarified adding more description of the surface renewal method background.* Added as requested.

*P5L138: Please add "ramps in the temperature signal caused by" between "geometry of" and "coherent structures"* Added as requested.

*P5L140 and L141: Please change to be clearer that the detection of the structure functions in the scalar signals improved.*

Added discussion of the improved detection and concurrent increase in statistical robustness with larger sample sizes (higher frequency measurement)

P5L144: Explain under which conditions.

Added as requested. Other reviewers noted the ambiguity of this statement so a short discussion of flux direction and atmospheric stability was added.

*P5L146: Please replace "to" by "for" if I understand well. In addition, explain why are the periods of 8 hours collected, and not continuously for two months. What challenges did you find? Was it only data collection for the unstable conditions?*

This language was too abbreviated in the text, and has been updated to reflect that I am describing several different experiments. The description now indicates that this data was used to test the methods with different performance criteria- i.e. data of different sizes, experiment durations, and captured at different frequencies.

*P5L147-L150: Please explain how this analysis is useful. Are fluxes better to be calculated for 3 minute periods?*

The introduction and discussion have been expanded to indicate why short duration fluxes are under consideration.

Atmos. Meas. Tech. Discussion doi:10.5194/amt-2017-123

*P5L163: What "total data size" means? It works with the same efficiency over 20Hz and 100Hz dataset? Please clarify.*

I have attempted to clarify this characterization of "total data size".

*P5L169-170: Is this the result of the authors' own research? If not, please give a reference.* I have gone back through my results and observations and changed this statement. I believe that my original intent was to indicate the relationship between friction velocity, vertical flux rate, and ramps times, and mis-stated these observations as being related to stability. I have changed the text and added a short explanation of the rationale, which is not a novel observation (i.e. higher velocity means faster exchange rates).

*P6L174: "again the largest factor in the difference between two methods" is making this phrase unclear. Please change to clarify* Changed for clarity.

**TECHNICAL CORRECTIONS:**

P1L10: One extra space seems to be typed in between the words "demonstrate" and "that"
P1L26: I think "manifests" is necessary instead of "manifest"
P2L60: "on" instead of "one"
P3L95: "an" instead of "a" before "application"
P4L111: Please use different word instead of "conditioning" if possible (i.e. "despiking")
P4L135: Please replace "product of" instead "product by"?
P5L165 and 166: are empty.
Avoid "@" in Figure titles.
These technical corrections are appreciated and have been addressed.

P4L135: Please replace "sense" by "importance in the flux measurement" or similar. Also, replace "generate" by "are"

The language has been changed for clarity. In particular, the word "sense" was used to indicate vector direction.

*P5L158: Please replace "difference vectors" by "the vectors of the differences". In addition, change "for each sample lag" by "for the corresponding sample lag"* This has been changed for clarity as suggested.

**General comments from Reviewer #3:**

These comments made by this reviewer have helped to make significant improvements in the revised manuscript. In particular, I was unaware of the review of despiking methods by Starkenburg et al. (2016), and this citation has significantly improved the context of this research and has helped narrow the discussion of the intent of my manuscript, and clarify what the novel aspects of the methods described here. Also, the reviewers prompted me to re-read of French et al.'s 2012 publication using the SR method, and the discussion of lag time there has helped improve my discussion of practical aspects of computation. Following the reviewers suggestions, I have re-written the abstract, introduction, and conclusions sections to clarify the intent, context, and relevance of the three computational methods described in the manuscript.

**Specific Comments:**

**Abstract: Logical presentation in the abstract needs improvement... What is your thesis?**

I acknowledge the lack of clarity and have re-written the abstract to clarify that the intent of the paper is to document and quantify the efficiency of three algorithms, which rely on convolution and algebraic simplifications, and consequently facilitate implementation of the surface renewal method. A secondary finding of the manuscript relates the flux averaging period and computational efficiency. Mention of potential uses for this findings (open source methods, implementing SR on mobile platforms, integration into hardware solutions, etc.) and similar have been moved to the discussion in the conclusions section, rather than being misleadingly referenced in the abstract.

**Not mentioned, but should be, are the SR limitations relative to EC... quantify the benefits: how much less does SR cost?**

I have added discussion relating more context and citations regarding practical implementation of SR to measure flux.

**What fetch benefit does one gain with SR?**

Because it is relevant to the context and motivation for the SR method generally, I have added language to better cite relevant references (Castellví 2012, Göckede et al. 2004; Paw U 2005), and brief discussion regarding the measurement fetch. I have kept this limited as it is not directly relevant to the computational method. In one sense, the method's reduction of measurement fetch is not clearly established in the literature, and following Castellví (2012), it is probably comparable to eddy covariance. However, it is clearly established that SR can give reliable measurements of flux within the roughness sub-layer (Paw U et al. 1995; Katul et al. 1996; Chen et al. 1997; among others more recently). Due to this difference from eddy covariance and gradient methods which require measurements at two heights, the practical source-sink area associated with a measured flux is smaller, and spatial footprints can therefore be resolved at finer scales.

L43: a major problem here: you have yet to publish your manuscript assessing efficacy of different averaging periods, but as of now we don't know the result and so do not know how important your algorithm refinement is.

I have expanded the description of the field experiments, and included results that quantify the improved calculation efficiency.

**Regarding de-spiking: authors should read and consider findings in Starkenburg et al. 2016, with attention to their Table 1.**

I am grateful for the reviewer for bringing this paper to my attention as it was published after this manuscript was originally prepared, and significantly adds to the context of the work shown here. While de-spiking is a required QAQC process with micromet data, the convolution method shown here specifically addresses and improves computational efficiency, and shows significantly improvement over the methods shown in Starkenburg's review of published methods. In addition, the convolution approach facilitates a more sophisticated application of the phase space approach, and overcomes a major limitation of that approach through efficient calculation. I have added a brief discussion of my method to the manuscript to demonstrate the benefit of signal processing techniques like convolution.

Discussion of convolution should mention that the time-series operation includes multiplication of the time-reversed kernel.

I have added more technical description of convolution.

Regarding structure function lag estimation, consider findings reported in French et al. discussing lag vs. SR accuracy.

This citation was also a good suggestion, and I have added a brief discussion of the relevance of lag time to the methods' accuracy, and how this pertains to computational efficiency.

**Root solving is not innovative although I will grant innovation in the diagnostic findings of pathological cubic equations by Edwards and Beaver. It would be useful to compare the speed improvement using Cardano's method vs. numerical Newton-Raphson.**

Although I grant that this would be an interesting comparison, I think that it suffices to state that algebraic solutions are always less computational intensive than any iterative root finding algorithm. Cardano's algebraic solution requires fewer operations than any numerical root findinig solution of which I am aware- it is a happy coincidence that he so-called "depressed" cubic polynomial is the one needed to solve the structure function arrangement posed by Van Atta.

**If your aim is open-source you should provide readers with evidence that m script successfully executes on a non-proprietary software platform.**

I agree! In the interest of time, I have not included my partial results of executing these methods in Python, which are ongoing. I rephrased this mention of potential open source implementations to my discussion as a needed action.

**Line 166: here is a critical finding about computation time vs. averaging period; you should find a way to highlight this and not bury it in the text.**

I agree that this is one of the coherent points in the results, and I have reworded to emphasize it.

**Computational Efficiency for the Surface Renewal Method**

Jason Kelley∗ and Chad Higgins

Dept. of Biological and Ecological Engineering, Oregon State University 116 Gilmore Hall, Corvallis, OR 97333 USA

5 \* Corresponding Correspondence to: Jason Kelley (Author, email kelleyja@oregonstate.edu)

**Abstract.**

Measuring surface fluxes using the surface renewal (SR) method requires a set of programmatic functions algorithms for tabulation, algebraic calculation, and data quality control. A number of different methods have been published describing automated calibration of SR parameters. Field experiments conducted over a variety of surface

- 10 conditions used eddy covariance (EC) as a control flux determination for sensible heat flux and evaporation. Because the SR method requires –utilizes high frequency (10 Hz+) measurements of a temperature or scalar traceconcentration, some steps in the flux calculation these tasks are computationally expensive, especially when automating SR to. Some implementations of SR, such as Experimental hypothesis optimization of averaging time, that require perform many iterations analysis of time averaging require many iterations of these calculations. For a
- 15 study on the minimum time to measure flux, To economize this type of analysis, ss Several new programmatic algorithms methods were written to that perform the required calculations more efficiently and rapidly, and tested for sensitivity to length of flux averaging period, ability to measure over a large range of lag time scales, and for overall computational efficiency. These algorithms, utilize signal processing techniques and algebraic simplifications that -demonstrateing simple modifications of thattomethods dramatically improve computational
- 20

simplifications that -demonstrate<del>ing</del> simple modifications <del>of</del>that<del>tomethods</del> dramatically improve computational efficiency. The results here complement efforts by other authors to standardize a robust and accurate computational SR method. Increased<del>Programming techniques</del> speed of computation time <del>such as these can</del>-grants flexibility to implementing the SR method, opening new avenues for SR to be <del>applied</del>used in research, for applied monitoring, and in novel field deployments.

**1 BackgroundIntroduction**

Originally described by Van Atta (1977), the SR model measures vertical flux that occurs during rapid events which manifest as coherent structures in a turbulent flow. The physical mechanisms are statistically distinct from those described in the Eddy-eddy covariance (EC) method, which has been established as -a robust and accurate method to measure flux (Baldocchi, 2014)(Baldocchi, 2014)., yet tThe surface renewal (SR-) method offers several advantages and complements over the use of EC to measure flux. While EC requires a high frequency fast (10 Hz+) measurement of both the vertical wind speed and air temperature to measure the sensible heat flux, the SR method does not explicitly require vertical wind speed, allowing can make this flux to be determined solely determination using from rapid measurements of temperature or other scalar concentrations air temperature measurement only. By ing ecause fewer, lower costed sensors are required, cost is d, which expands the SR method theoretically can be

used for directflux measurement from research applications to more general to develop the SR method for field

- 35 applied<del>cations</del> monitoring (Paw U et al., 2005; Spano et al., 2000). Another advantage of SR is over methods such as EC Previous studies have shown isthat the ability to measure flux measurements can be captured very near the surface or near the top of the plant canopy using SR-(Katul et al., 1996; Paw U et al., 1992)(Katul et al., 1996; Paw U et al., 1992). By taking measurements very close to the surface, the measurement fetch is effectively reduced and consequently the effective so called "area from which the flux can be attributed footprint" or contributing area is 40 smaller (Castellví, 2012),- yielding a more localized flux estimate.

The SR method estimates turbulent transport rates from fast response measurements of scalar properties such as temperature or trace gas concentration. In the SR conceptual model, rapid changes in scalar concentration are associated with episodic displacement of near-surface air parcels, and the surface condition is renewed from upper air. While in proximity to the surface, the air parcels are gradually enriched or depleted in temperature or scalar 45 concentration by diffusion (Castellví et al., 2002; Paw U et al., 1995). The majority of flux from the surface is attributed to these rapid ejections, which distinguish coherent structures in near surface atmospheric motions (Gao et al., 1989). The duration and amplitude of these rapid fluctuations (visible as ramps in the scalar trace) are used to

determine the magnitude and direction of the flux density. - As in Taylor's concept of frozen turbulencedescribed by, small scale turbulent transport primarily occurs in during rapid events which manifest as (ramp like coherent

- 50 structures). Consequently, ipotentially t should anbe possible to measureestimate flux over shorter periods with SR than is typical for EC (15 30 minutes). One possible use of this kind of rapid flux measurement will facilitatewould be to map spatially heterogeneous flux using, near surfacelightweight sensors mounted on a mobile device. A mobile application of SR could provide missing information for understanding sub basin scale hydrology, to validate downscaled models, and facilitate efficient water use throughheterogeneity precision irrigation systems.
- 55 Another advantage is by reducing sensor cost, SR can allow more extensive flux measurements than can be accomplished with expensive equipment such as EC systems. By measuring extensively, the spatial heterogeneity of surface fluxes can be explored, and adding to the data collection capacity of sparsely located, tregional weather networks. Low cost sensors also facilitate measuring flux in situations where deploying more expensive equipment would be infeasible, such as direct monitoring of farm water use, at unsecured field sites, and in developing
- 60 regions.standardization of calibration and quality control measures to establish , uniform, and te methods Because of the short duration of these events, the SR method complements spectral methods to evaluate the flux contributions made over time scales shorter than the typical 15-30 minute averaging time used for EC (Katul et al., 2006; Shapland et al., 2012a, 2012b). Rapid flux measurement will facilitate new applications, such as spatial mapping of flux using vehicle mounted, near-surface sensors, and real-time monitoring systems. Mobile SR 65
- implementations and other novel field methods could provide new insights into the complexities of sub-basin scale hydrology, be used to validate downscaled models, and measure the heterogeneity of flux at sub-field scales. The implementation of specific The use of SR to map heterogeneous surface flux fromonto map flux requires a prescribed will first will require establishing knowing a the minimum averaging time period (on the order of minutes) and ramp time duration (on the order of seconds), in which SR can resolve for robustinfor which a

70 representative and statistically robust a flux magnitude can be determined can be obtained robustly. To implement SR on a moving vehicle (for instance, to map spatially variable flux), finding a minimum averaging time is desirable to increase the spatial resolution of the resulting map. The averaging time and lag time used in the SR method relate the sensitivity of the scalar measurement to the time scales at which most significant flux occurs (Shapland et al., 2014). To find the minimum required measurement period, several field studies were conducted over in 2014 and -

- 2015 over various types of surface conditions. This required a rapid computational method that worked F, and 75 fluxes were computed over a range of different time averaging periods, and which could implement the various calibration procedures used in the SR method-with co-located EC and SR sensors, with several different surface, weather, and crop conditions. Initial attempts to calculate flux Initially, the computation fF followededing the methods as described by described in Paw U et al. (2005a) and Snyder et al. (2008) literally. However, 80 implementing these methods as documented but-wasere hampered by slowed by-computation time, which constrained for the processing many required iterations required to determine the minimum flux averaging period Paw U et al., 2005 and Snyder et al., 1996. However, executing these methods literally as described was constrained by computationfound to be inefficient time.
- As late as the 1990's, limits on computing power, data logger memory, and telemetry limited the implementation of 85 SR to highly skilled researchers (Katul et al., 1996; Snyder et al., 1996). Open source software and online forums are abound with methods that utilize advances in computing power, memory availability, and the accessibility of multithreaded processingwhen calculating over the many iterations that were required for multiple averaging periods. - More efficient methods were found to utilize the increased computing power that is currently available, and to take advantage of parallel processing approaches. These approaches methods reduce computational
- 90 overhead, and open possibilities can augment the -for-SR technique as-to allow implementation -which can be a constraint when with using low cost data computers and data loggers, or where remote telemetry is required. TThree example methods are shown here are shown which elarify streamline specific operational steps in the SR method. The first is a rapid-method adapted from signal processing to "despike" noisy data, a quality control technique commonly required-used to-in processing raw meteorological data. Second is a method to compute
- 95 structure functions over multiple time lags rapidly using convolution in two dimensions. Third, an algebraic solution array calculation is used to to find the root find the -of a cubic polynomials roots used to , which facilitates the rapid-determinen the SR ation of SR-ramp amplitude as an array calculation. By using more efficient algorithms, rapid iterative trials can be conducted to adjust calibration parameters, test hypotheses on the time averaging of flux calculations, and potentially measure SR flux in real-time.
- 100 Advantages such as low cost, relatively simple instrumentation, and easier field implementation are all cited as motivating factors to use the SR method (Paw U et al., 2005), yet work remains to standardize a robust method (French et al., 2012; Suvočarev et al., 2014). Because sensor cost is reduced, SR systems can be implemented to measure flux more extensively than EC, and in situations where EC is impractical. Extensive, site specific SR estimates can augment the utility of sparsely located, permanent weather stations in mapping the heterogeneity of 105 surface flux. Examples of situations which could benefit from low cost flux measurements include direct crop ET

monitoring, experiments at remote field sites, and in developing regions. While SR may expand flux measurement

applications, the method still requires standardized calibration and quality control measures to establish that SR is robust and accurate, and a critical step in developing the method is to reduce computation costs.

**110 2 Methods**

The examplese computational approaches algorithms shown here were adaptimprove or economize existinged from calculation previously described methods, including despiking of scalar tracestime series data (Højstrup, 1993; Starkenburg et al., 2016)(Højstrup, 1993)(Vickers and Mahrt, 1997), calculation of structure functions (Antonia and Van Atta, 1978)(Antonia and Van Atta, 1978), and Fourier analysis of signals i.e. spectral analysis (Press, 2007; 115 Stull, 1988)(Stull, 1988). In each case, dramatically faster execution times were accomplished using simple programming improvements. Most efficiency gains were due a result of code to "vectorization", which is the conversion ofting iterative looping algorithms from an iterative loop process-into an array calculations. All methods described -here were implemented in the Matlab language (The Mathworks Inc., 2016)(The Mathworks Inc., 2016), with including the Statistics, Curve Fitting, and Signal Analysis Toolboxes. Matlab's Profiler (profile.m) was used 120 to track the memory demand and speed of time to implement calculations ation. Trials were conducted on multiple desktop systems; for uniformity, analysis shown here data from only used Test test runs shown her used in this analysise that were conducted one one system, a in the Windows 10 operating system running on using an Intel Core (TM) i7-3720QM processor operating at 2.60GHz with 16GB RAM. Processor clock speed was verified using the Matlab's Profiler tool at run time, and all-processing times reported here include are described as both-run time 125 (actual observed execution time) and or asa Total Run Time, which is the sum of CPU time used in for all calculation threads. Example methods are indicated by function name (in italics)., and full Abbreviated, commented eode scripts for the example functions is are provided in the supplementary material. Complete code for this analysis can be obtained directly from the corresponding author. Although these methods were prototyped in Matlab, the examples shown are generally useful as solutions to challenges commonly encountered in 130 micrometeorology. Example data was collected during two fvarious field experiments in-from 2014 and 2017,5 using an integrated sonic anemometer and infrared gas analyzer (IRGASON) and fine wire thermocouples (FWTs) which was, and was recorded at 10, Hz and 20, and 100 Hz using a CR1000 datalogger (Campbell Scientific). The data used in verifying the methods is provided through supplementary materials online (DOI: 10.7267/N9X34VDS).

135 Vector and array calculations are more efficiently executed than iterative methods; Vectorization-vectorization of Matlab code entails removing loops (which are not pre-compiled) and taking advantage of implicit parallel methods in Matlab's pre-compiled library-functions (Altman, 2015)(Altman, 2015). Other significant improvements were enabled through the Fast Fourier Transform by using convolution of number arrays, rather than iterative operations. In the case of determining ramp geometry in the SR method, Cardano's solution for depressed cubic polynomials
140 (published in 1545) reduces a root finding algorithm from an iterative numerical approximation to an algebraic exact algebraic vector vector calculation. While some of the implementation of these se programming-methods address unique aspects-are particular to the Matlab language, the general mathematical concepts are universal. Although

these methods were prototyped in Matlab, the examples shown are generally useful as solutions to challenges commonly encountered in micrometeorology. By reducing computational overhead, these methods expand the possible uses of SR flux measurements to monitor agricultural water use, to measure from lightweight aerial platforms, and in other specialty applications.

145

**Example 2.1: Despiking of noisy data using convolution**

150

procedure when measuring environmental parameters, such as wind speed, temperature, and trace gas concentrationespecially in challenging conditions or complex environments (Göckede et al., 2004; Starkenburg et al., 2016)(Göckede et al., 2004). The origin of spikes in a time series can may be or electronic be either physical (sensor malfunction or actual physical non-errors) or electronic. in nature; regardlessRegardless of the origin, these spikes can be recorded as either abnormally large or small values, or may be marked by as some a pre-firmware defined error flag. Spikes become problematic which is fif they are not readily differentiated when during automatic

Despiking is the removal of erroneous or extreme data points from a time series of sampled values. It is a common

- 155 data automatically-importsing data -(Rebmann et al., 2012)(Rebmann et al., 2012). Spikes interfere with normal statistical calculations,, and requiring-require some deliberate and objective method to identify, remove, and interpolate where they exist. For instance, a data logger program may record an error as "9999" or a character string, while Matlab denotes missing values in a numerical array as NaN ("not a number"). Because normally distributed data may contains noise across-in a wide range of scalesvalues, robustness of the despiking algorithm is
- 160 complicated by the requirement to differentiate between "hard spikes" characteristic of automatic flags (such as 9999) and "soft spikes" which are realistically valued, but exceed some objective limiterroneous measurement. This An objective limit for soft spikes is usually defined as appropriate for the signal to noise ratio of any particular data, usually in terms of variance during some-a stationary-defined windowing period. Clearly distinguishing between these explicit errors and flagged errors can be achieved by a static objective criteria, by a dynamic statistic, or in a separate pre-processing operation. Previous authors have described a variety of methods including use of
- autocorrelation (Højstrup, 1993)<del>(Højstrup, 1993)(Højstrup, 1993)</del> and statistics within a moving window (Vickers and Mahrt, 1997)<del>(Vickers and Mahrt, 1997)</del>. A comprehensive review of despiking methods is presented by Starkenburg et al., (2016), with emphasis on the accuracy and statistical robustness of different computation methods.
- Despiking is largely a problem of conditional low pass filtering; consequently this procedure can be treated as an application of signal conditioning-processing and-which can be therefore performed efficiently using convolution. Mathematically, convolution can be understood as a multiplicative function that combines a data signal with a filter signal. TFor a discrete signal, the filter is a weighting array , the size of the moving window, which is multiplied (in the Fourier domain) with data inside the a window. In the time domain, the window can be visualized whichas it is movinged along the data array as it is multiplied. For As examples, a window-filter with a weight of 2 at the center of the window, and zeros elsewhere, would amplify the data signal by a factor of two;- Caonvolution with a

the data-over 10 samples. The computational efficiency of convolution is a result-product of the Fast Fourier

window filter 10 units samples wide,, each weighted at 0.1, generates a 10 samples running average (smoothed) of

Transform (FFT), which allocates memory efficiently by a process known as bit switching. An thorough treatment

- 180 of bit switching can be found in -(Chapters 12 and -13, in of Press (2007)Press, 2007). To demonstrate the increased efficiency of the FFT, two programs methods were used to despike 8.5 hours of 20 Hz sonic temperature data (609139 samples). A first attempt One method utilized a sliding window in a for loop, following the objective criteria described by Vickers and Mahrt (1997). AThen improved second program-method (shown in despike.m) used convolution to determine a running mean and standard deviation used in the identification of spikes. After
- 185 multiple runs with different input criteria, the first program average run time was 27 seconds. Using convolution, the second program average run time was 0.2 seconds, decreasing run time by approximately 99%. While this drastic improvement may potentially overemphasize the slow compile times of for loops in Matlab (compared to other languages), it nonetheless demonstrates the value of the FFT in speeding calculations with time domain signal conditioning signals. Faster processing time - and facilitatesd the abandonment of time saving strategies used in
- 190 earlier applications of despiking, such as skipping samples or limiting window sizemore comprehensive, calibrated, and accurate analysis, and can reduce data loss compared to coarser filtering techniques. To measure record test computation time independently of hardware uniformly (processor speed and RAM size), identical 10 Hz data was sub-sampled in-to record lengths from of 0.25 to 48 hours, and multiple runs were despiked with each sample set. Raw data was checked for hard error flags which required text to number conversion, but was
- 195 not otherwise manipulated prior to despiking. Matlab Profiler was used to track the run time for all threads, using the undocumented flag "built-in" to track pre-compiled Matlab functions as well as <del>custom-user</del> functions1. The total run time for all threads was tabulated and averaged across sets of each data length (Figure 1). Despiking in despike.m was accomplished with convolution used to calculate the running mean and running standard deviation arrays, which are used as comparators in identifying spikes (see supplementary materials). By using convolution,
- 200 despiking was two orders of magnitude faster for all lengths of data. To illustrate the effect of Matlab's built-in parallel processes, Figure 2 shows the ratio of actual run time to Total Run Time, indicating that the convolution method relies on computations conducted in parallel for processing increasingly larger-longer data records. This benefit is direct accrued from the efficiency of the FFT.
- With increased computation speed, automatic and accurate despiking can be accomplished, with reduced time cost 205 to determine any necessary calibrate for the procedure. The various methods employed to despike data are variously limited by computational inefficiency (Starkenburg et al., 2016). "Phase space thresholding", originally described by Goring and Nikora (2002) is one such method that Starkenburg noted as being hampered by computational costs, and by a requirement for iterative applications to calibrate despiking parameters. By decreasing the execution time, a similar method was developed that allows rapid and accurate despiking of data, both for the detection of hard and
- 210 soft spikes. A phase space method allows objective criteria to be calibrated for specific sensor data, and a visual diagnostic phase space diagram that allows for rapid calibration of the despiking criteria (Figure 3). Projecting the signal into a phase space diagram reveals modes related to sensor error, response time, and other factors leading to spikes. Using convolution to determine moving window statistics (such as a moving mean, standard deviation, etc.), objective identification of behaviors characteristic to a particular sensor response. In figure 3, infrared gas analyzer

<sup>1 http://undocumentedmatlab.com/blog/undocumented-profiler-options-part-4, Accessed January 2017

215 data (in this case, signal strength) collected at 20Hz for 17 days is projected with one minute moving window statistics. Based on this projection, a cut-off in phase space for spike identification can be assigned, and the subsequent percentage of removed data calculated. In this case, the sensor was repeatedly affected by dust from farm operations (Figure 4), yet only 1.5% of the data was required to be removed as spikes due to the precision of the despiking algorithm. This procedure took less than five seconds of computation.

Figure 1: The total computation time is the sum of CPU time spent on all calculation threads. Marker is the mean run time for multiple runs, which varied from 30 runs (15min – 4h data) to 10 runs (8, 12, 24 h). 48h calculation is represented by one run only. Error bars represent one standard deviation of all runs.

Figure 2: In addition to faster overall run times, calculation of larger data sets remains fast in part due to implicit parallel processing via the Fast Fourier Transform, which can be readily conducted on multiple threads. Parallel processing of noise in the loop method is not apparent.

**220 **Example 2.2:** -Structure function calculation**

Another computationally- intensive process in SR is the determination of the 2nd, 3rd, and 5th order structure functions. Ramps are an identifiable feature in the measured temperature trace above any natural surface, yet calculating determining the the amplitude of characteristic ramp geometrys from high frequency data requires an

efficient, robust, and preferably automated procedure. There are several methods to determine ramp geometry, including visual detection (Shaw and Gao, 1989)(Shaw and Gao, 1989), low pass filtering (Katul et al., 1996; Paw U et al., 1995)(Katul et al., 1996; Paw U et al., 1995), wavelet analysis (Gao and Li, 1993)(Gao and Li, 1993), and structure functions (Spano et al., 1997)(Spano et al., 1997). Structure functions in particular provide both objective criteria to detect ramps and an efficient method to tabulate ramp geometrystatistics of time series data, and use of structure functions has become the predominant method used for SR. The general form for a structure functions is:

$$S^{n}(r) = \frac{1}{N-r} \sum_{i=N-r}^{i=N-r} \left[ T(i+r) - T(i) \right]^{n} - \frac{Equation}{(1)}$$

- 230 in which a vector of length N-1 is composed of the differences between sequential (Temperature) samples T(i), separated by lag r. The structure function  $S^n(r)$  of order n for a given sample time-lag r is obtained by finding differences between each sample i and sample i r (lagged in time by r samples). The nth order structure function are defined by the nth power in the summation raising the difference vector to the **n** power, summing the vector and normalizing by N-1. In a turbulent flow field, The the sampled fluctuations of scalar time series -of signal  $\Delta T(i)$ - in 235 a turbulent flow can be shown to beare a combination of random fluctuations and coherent structures (Van Atta and Park, 1972)(Van Atta and Park, 1972). The random (incoherent) part of the signal is produced a product by of isotropic turbulent processes, and over an a-statistically significant dequately large sample this sample should thus haves no particular directional sense or orientation (by the isotropic definition). On the other hand, By contrast, coherent structures are by definitiongenerate characteristic anisotropic anisotropic signatures, with periods of 240 gradually rischange punctuated by period and sharp transitions. These sharp transitions occur during "sweeps and ejections" of parcels enriched or depleted in scalar concentration (heat or water trace gas), evidence of transport from an Eulerian perspective. The total variance is composed of additive random and coherent components, and sStructures functions can be used to decompose the measured tracetime series fluctuations into random and
- 245

Advances in sensor response time and processor speed have revealed an increasingly detailed picture of the coherent ramp structures. In deriving a method to find ramp geometry, Van Atta (1977) calculated structure functions for eight different lags. Two decades later, increased processor power and memory size allowed Snyder et al., (1996) to calculate structure functions on 8 Hz data for lags from 0.25-1.0 seconds, but they were unable to resolve fluxes

coherentisotropic and anisotropic components and identify the characteristic ramp amplitude and duration of

coherent structures (Van Atta, 1977)(Van Atta, 1977).

accurately at some measurement heights and surface roughness conditions. Later it was realized that determining the contributions from "imperfect ramp geometry" would require more thorough examination of ramp durations (Chen et al., 1997a; Paw U et al., 2005)(Chen et al., 1997; Paw U et al., 2005a).

For the this analysis-of minimum SR averaging periods, data was used from several field experiments. The data records used collected ranged in length over from long measurement periods (8 -24-hours) to over two months, with

255 at sampling frequencies of 10Hz, 20Hz, and up to 100Hz (fastest frequency for short duration trials only). Initially, Computation computation of structure functions with the first method (series of nested *for* loops) for 3 minute periods with lags up to 10 seconds required an average 39 seconds computation timeon average with a series of nested loops. In contrast, using 2-d-the convolution method, accomplishes this same calculation was accomplished

285

in 7.6 seconds,; an ~80% reduction in execution time. The function *strfnc.m* (provided in *Supplementary Materials, S1*) also simultaneously time stamps the averaging period, finds the sign of  $S^3(r)$  (used to find flux direction), and indexes the maximized value of  $S^3(r)/r$ , preparing the data for subsequent steps in determining flux. Using 100 Hz FWT data increased the processing time using the convolution method to 38.4 seconds. , demonstrating that tThe loop based-method would be too slow fo unable tor processing 100 Hz data with large data sets or in real time applications, and would require long calculation time when using large continuous data records.

- For a total of N sample lags, Two-two dimensional convolution is accomplished-performed with-using a filter matrix, which is composed of ith with each N column vectors of length N+1: s as [1 -1 0 0...0], [1 0 -1 0 ...0],...[1 0 0 0.... -1]0 0 .... 1 ...]., -eEach column in the filter-representsing amatrix representing a time sample lag increasing distance. When the filter matrix is convolved with time series data, The the resulting column matrix is composed of vectors of the resulting matrix are -difference vectors, each representing the difference of the element-wise
- 270 differences (T(i+r) T(i)) as in Eq. 1; these vectors correspond to each sample lag in the at each sequential lag for *r* (Equation 1) filter. Trials of 10Hz data using Matlab's Profiler showsed that calculation efficiency is not accrued directly from convolution, but by changing the order of implementation—when multiplying large arrays (for exponentials) and summing. In the looping method, the 2,3, and 5th exponentiation (n = 2,3,5) is conducted on the difference vectors for each lag separately. The Exponentiation is implemented accelerated exponentiation in the
- 275 convolution method viain *strfnc.m* is possible by using matrix multiplication of large matrices on the convolved matrix, which and is possible faster once the data is reordered bydue to compact memory allocation of the FFT-2 d convolution. The resulting efficiency (calculation time for a given data size) is not doesn't dependant on total data sizesize, but is strongly dependent on the length of the averaging period used to calculate flux partition the data (Figure 5)-(Figure 3). In other words, the choice of averaging period length is the most significant factor in
- 280 computation time of the maximized structure functions used to determine ramp geometry. Computation time increases rapidly for periods shorter than 5 minutes. With the ultimate goal of investigating the shortest robust SR measurement period The length of averaging time, this proves isto be a critical improvement consideration initerative SR calculations used to developing a rapide SR measurement method.

In most SR studies to determine flux, a lag time is assigned to the structure function calculation, with only a few authors allowing for a procedure to maximize the ratio  $S^{3}(r)/r$  (Shapland et al., 2014). Yet lag time has been